# In Vivo Phosphorylation of the Cytosolic Glucose-6-Phosphate Dehydrogenase Isozyme G6PD6 in Phosphate-Resupplied *Arabidopsis thaliana* Suspension Cells and Seedlings

**DOI:** 10.3390/plants13010031

**Published:** 2023-12-21

**Authors:** Milena A. Smith, Kirsten H. Benidickson, William C. Plaxton

**Affiliations:** Department of Biology, Queen’s University, Kingston, ON K7L 3N6, Canada; 17mas6@queensu.ca (M.A.S.); 17kb29@queensu.ca (K.H.B.)

**Keywords:** CDPK, glucose-6-phosphate dehydrogenase, oxidative pentose–phosphate pathway, protein phosphorylation, phosphate nutrition, post-translational modification

## Abstract

Glucose-6-phosphate dehydrogenase (G6PD) catalyzes the first committed step of the oxidative pentose phosphate pathway (OPPP). Our recent phosphoproteomics study revealed that the cytosolic G6PD6 isozyme became hyperphosphorylated at Ser12, Thr13 and Ser18, 48 h following phosphate (Pi) resupply to Pi-starved (–Pi) *Arabidopsis thaliana* cell cultures. The aim of the present study was to assess whether G6PD6 phosphorylation also occurs in shoots or roots following Pi resupply to –Pi *Arabidopsis* seedlings, and to investigate its relationship with G6PD activity. Interrogation of phosphoproteomic databases indicated that N-terminal, multi-site phosphorylation of G6PD6 and its orthologs is quite prevalent. However, the functions of these phosphorylation events remain unknown. Immunoblotting with an anti-(pSer18 phosphosite-specific G6PD6) antibody confirmed that G6PD6 from Pi-resupplied, but not –Pi, *Arabidopsis* cell cultures or seedlings (i.e., roots) was phosphorylated at Ser18; this correlated with a significant increase in extractable G6PD activity, and biomass accumulation. Peptide kinase assays of Pi-resupplied cell culture extracts indicated that G6PD6 phosphorylation at Ser18 is catalyzed by a Ca^2+^-dependent protein kinase (CDPK), which correlates with the ‘CDPK-like’ targeting motif that flanks Ser18. Our results support the hypothesis that N-terminal phosphorylation activates G6PD6 to enhance OPPP flux and thus the production of reducing power (i.e., NADPH) and C-skeletons needed to establish the rapid resumption of growth that ensures Pi-resupply to –Pi *Arabidopsis*.

## 1. Introduction

Glucose-6-phosphate dehydrogenase (G6PD) catalyzes the first committed step of the oxidative pentose–phosphate pathway (OPPP) by irreversibly converting glucose-6-phosphate (G6P) and NADP^+^ into 6-phosphogluconolactone and NADPH. The OPPP’s principal functions are to produce: (i) NADPH needed for ‘reductive’ biosynthetic reactions and reactive oxygen species (ROS) detoxification, and (ii) anabolic precursors (e.g., ribose-5-phosphate and erythrose-4-phosphate) required for nucleotide, aromatic amino acid, and co-factor synthesis (Figure 1) [1].

Plant G6PDs are derived from a small multigene family that encode distinct cytosolic and plastidic isozymes (G6PDc and G6PDp, respectively) that exhibit marked differences in their respective molecular and kinetic properties. G6PDp isozymes are primarily expressed in green tissues, where they are reversibly inactivated in a light-dependent manner by a thioredoxin-mediated, dithiol–disulfide interconversion mechanism that helps to maintain photosynthetic efficiency [1]. G6PDc is believed to have a more central role in regulating basal metabolism, as it contributes the majority (~80%) of G6PD activity in plant tissues [2,3]. However, the functions and regulation of G6PDc are less understood relative to G6PDp. G6PDc isozymes lack the conserved redox-sensitive cysteine residues found in G6PDp and appear to be primarily controlled by the intracellular NADPH/NADP^+^ ratio [4,5]. The *Arabidopsis G6PD* gene family encodes four plastidic (G6PD1-4) and two cytosolic (G6PD5 and G6PD6) isozymes, as indicated by genome annotation and transit peptide sequence analysis [4]. Interestingly, alternative splice variants of G6PD5 localize at the cytosolic face of the endoplasmic reticulum, along with other metabolically sequential OPPP enzymes (i.e., 6-phosphogluconolactonase and 6-phosphogluconate (6PG) dehydrogenase) [6]. These endoplasmic-reticulum-associated G6PD5 isoforms likely operate within a membrane-bound metabolon to provide NADPH for various metabolic pathways under stressful conditions that require enhanced NADPH production via the OPPP.

Apart from supporting anabolism and cell growth, G6PDc and the OPPP play a crucial role in mitigating oxidative stress through the generation of NADPH, an essential co-factor of cellular antioxidant systems [1]. For example, to attenuate salinity-induced oxidative damage, salt-stressed plants require an enhanced supply of NADPH for osmolyte synthesis, ROS detoxification, and the maintenance of ionic balance [7]. G6PDc has been identified as a key player during the acclimation of wheat, barley, soybean, and *Arabidopsis thaliana* to salinity stress [7,8,9,10]. Salt-stressed *Arabidopsis* seedlings lacking G6PDc (i.e., *g6pd5/6* double mutants) displayed reduced root growth, low germination rates and enhanced accumulation of ROS [11]. Moreover, the salt-stress-induced expression of ascorbate peroxidase and glutathione reductase was attenuated in the *atg6pd5/6* mutant, suggesting that G6PDc plays a key role in mediating ROS scavenging during salinity stress [11]. It is also notable that *g6pd5/6* knockout plants were more sensitive to salt stress than the corresponding single mutants, which displayed only slight phenotypic impairments [11]. Nevertheless, the high abundance of *G6PD*6 mRNA and in vivo activity throughout the plant indicates that G6PD6 is the predominant G6PDc isozyme expressed in *Arabidopsis* [4]. G6PD5 and G6PD6 are likely coordinated in their role of cytosolic NADPH production within various *Arabidopsis* tissues, and appear to be post-transcriptionally regulated by an unknown mechanism, given the low correlation between *G6PD5* and *G6PD6* mRNA abundance and corresponding G6PD activities or amounts [4,12].

Inorganic phosphate (Pi; HPO_4_^2−^) is an indispensable but environmentally limiting macronutrient that has crucial functions in energy transduction and metabolic regulation while also serving as a structural constituent of essential biomolecules such as nucleic acids, phospholipids, sugar-phosphates, and adenylates. Pi also controls the activity of many plant proteins via its covalent attachment to phosphoproteins and/or by acting as an allosteric activator or inhibitor of key regulatory enzymes of central metabolism [13]. However, Pi is one of the least available macronutrients in many terrestrial and aquatic ecosystems. Plants have therefore evolved an impressive array of adaptive strategies known as the ‘Pi starvation response’ (PSR) that facilitate their acclimation to the suboptimal levels of soluble Pi that characterize most soils. The PSR is regulated by Pi-starvation inducible genes, as well as various post-transcriptional and post-translational strategies that reprioritize internal Pi use while maximizing root Pi acquisition [14]. For example, –Pi plants optimize root Pi uptake via the following processes: (i) induction of high-affinity Pi transporters that actively assimilate Pi against a steep concentration gradient; and (ii) exudation of Pi-mobilizing compounds such as organic acid anions (e.g., malate, citrate), nucleases, and purple acid phosphatases [14]. Protein post-translational modifications (PTMs) such as phosphorylation and ubiquitination play a central role in plant PSR signal transduction pathways and the control of phosphate-starvation-inducible gene expression and gene products. For example, during Pi deprivation, root phosphoenolpyruvate carboxylase (PEPC), a tightly regulated cytosolic enzyme situated at a pivotal branchpoint of central metabolism, is activated by protein-kinase-mediated phosphorylation at a conserved N-terminal serine residue [15,16]. This enhances anaplerotic partitioning of imported sucrose into Pi-mobilizing organic acid anions [16,17].

We recently reported that proteomic remodeling occurring 48 h following the resupply of 2 mM Pi to heterotrophic, –Pi *Arabidopsis* cell cultures occurred largely at the level of protein phosphorylation, with limited fluctuations in protein abundance [18]. It is notable that G6PD6 was one of the most differentially phosphorylated of the over 500 proteins whose phosphorylation status significantly changed following Pi resupply. Multiple N-terminal residues of G6PD6 (i.e., Ser12, Thr13 and Ser18) were hyperphosphorylated following Pi resupply [18], highlighting a potential link between Pi nutrition and a novel post-translational G6PD6 regulatory mechanism via reversible phosphorylation. The initial aim of the present study was to interrogate phosphoproteomic datasets to assess the prevalence of N-terminal phosphorylation of G6PD6 and its orthologs. We then developed an anti-(pSer18 phosphosite-specific G6PD6) antibody (anti-pSer18) for immunoblot assessment of the impact of Pi nutrition on G6PD6’s in vivo Ser18 phosphorylation status in *Arabidopsis* cell cultures and seedlings. Our results are consistent with the hypothesis that this PTM activates G6PD6 to mediate increased cytosolic OPPP flux, and thus enhance the provision of NADPH and biosynthetic precursors needed to support the resumption of cell growth that rapidly occurs following Pi resupply.

## 2. Results

### 2.1. Multisite N-Terminal Phosphorylation of Arabidopsis G6PD6 and Its Orthologs Is Quite Prevalent

A multiple sequence alignment was conducted to evaluate the conservation of G6PD6’s Ser3, Ser12, Thr13, and Ser18 phosphosites [18,19,20] amongst the N-termini of other G6PDs (Figure 2). These residues are highly conserved between the two *Arabidopsis* G6PDc paralogs, G6PD5 and G6PD6 (≈90% sequence identity), as well as relatively well conserved between G6PDc orthologs from other vascular plant species (i.e., soybean, potato, tomato, barley, rice). By contrast, these sites are not evident in *Arabidopsis* G6PDp isozymes (i.e., G6PD1-4), which share much lower overall sequence identities with G6PD6 (≤45%) (Figure 2).

Interrogation of various phosphoproteomic datasets indicates that N-terminal phosphorylation of *Arabidopsis* G6PD6 and its orthologs is relatively common (Figure 2 and Table 1). Of the two *Arabidopsis* G6PDc isozymes, N-terminal phosphorylation of G6PD5 appears to occur less frequently (Table 1), implying that G6PD6 may be the primary G6PDc isozyme subjected to this PTM. Similar N-terminal phosphosites have also been identified in vertebrate (Ser8/Thr10 for human and mouse) and invertebrate (Ser7 for *Caenorhabditis elegans*) animal G6PDs (Figure 2) despite their phylogenetic distance from *Arabidopsis* G6PD6 (≤48% sequence identity).

### 2.2. Residues Flanking G6PD6’s N-Terminal Phosphosites Are Representative of a Basophilic Motif

Residues neighboring G6PD6’s N-terminal phosphosites comprise an S- or T-type motif [40], with basic (i.e., Lys, Arg or His) residues occurring at positions −1, −2, −5 and +3 for Ser12; −2, −3, −6 and +2 for Thr13; and −3, −7 and +3 for Ser18 (Figure 3). Basophilic motifs are commonly recognized by Ca^2+^-dependent protein kinases (CDPKs), SNF1-related protein kinases (SnRKs), and Ca^2+^/calmodulin-dependent protein kinases [40,41,42]. In particular, G6PD6’s N-terminal phosphosites present several motifs reminiscent of the optimal recognition motif for plant CDPKs and SnRKs, i.e., containing basic residues at positions −6, −3 and +5, and hydrophobic residues at positions −5 and +4 (Figure 3) [40,41,42]. N-terminal phosphosites of G6PDc isozymes from other plant species possess similar flanking residues (Figure 2), indicating a potentially conserved mechanism of regulation.

### 2.3. Structural Modeling of G6PD6 Indicates That Its N-Terminal Region Is Intrinsically Disordered

The *Arabidopsis G6PD6* gene encodes a 515 amino acid polypeptide with a predicted size of 59 kD. The first 22 N-terminal residues of G6PD6 appear to be intrinsically disordered as predicted via MobiDB and structural modeling (Figure 4).

### 2.4. Specificity of Anti-pSer18 Immune Serum

Rabbit-phosphosite-specific antiserum was raised against a synthetic phospho- (P-) peptide corresponding to residues 10–24 of G6PD6 with phosphorylated Ser18 (Figure 5A). pSer18 was selected (as opposed to pSer12 or pThr13) as (i) it is the most frequently documented P-site of G6PD6 (and its orthologs) in various phosphoproteomic studies (Table 1), and (ii) Mehta et al. [18] determined that G6PD6’s Ser18 residue exhibited the greatest log fold-increase in phosphorylation 48 h following Pi resupply to –Pi *Arabidopsis* cells.

Immunodot blotting of synthetic P- and dephospho- (deP-) peptides revealed that a 1:500 dilution of anti-pSer18 antiserum cross-reacted with as little as 10 ng of the P-peptide, but also weakly cross-reacted with higher amounts of the corresponding deP-peptide (Figure 5B). However, the inclusion of 10 µg/mL deP-peptide with the diluted antiserum abolished this non-specific cross-reaction. Therefore, all subsequent anti-pSer18 immunoblots were probed in the presence of 10 µg/mL deP-peptide unless otherwise indicated. By contrast, incubation with 10 µg/mL of blocking P-peptide eliminated any cross-reaction of anti-pSer18 with either P- or deP-peptide (Figure 5B).

### 2.5. G6PD6 Phosphorylation at Ser18 Is Influenced by Pi Nutrition in Arabidopsis Suspension Cells and Roots

The use of anti-pSer18 for examining G6PD6 phosphorylation via immunoblotting was complemented with rabbit antiserum raised against purified pea G6PDc (anti-pea G6PDc) [44]; pea G6PDc shares 81.4% sequence identity with *Arabidopsis* G6PD6. The anti-(pea G6PDc), which detects both P- and deP-G6PD6, allowed for the normalization of total G6PD6 on immunoblots. Clarified protein extracts of –Pi and 48 h Pi-resupplied *Arabidopsis* suspension cells and seedlings (i.e., roots and shoots) cultivated in sterile liquid media were subjected to SDS-PAGE followed by immunoblotting with anti-pSer18 and anti-(pea G6PDc) (Figure 6 and Appendix A).

Phosphorylation of 59 kD G6PD6 polypeptides at Ser18 was evident on immunoblots of Pi-resupplied cells and root extracts probed with anti-pSer18 in the presence of 10 µg/mL corresponding deP-peptide, with no apparent change in the abundance of 59 kD G6PD6 polypeptides (based upon anti-(pea G6PDc) immunoblotting) (Figure 6). By contrast, no cross-reaction of anti-pSer18 with 59 kD G6PD6 polypeptides occurred when immunoblots of the Pi-resupplied suspension cell or root extracts were probed in the presence of 10 µg/mL of the blocking P-peptide (Appendix A); this further attests to the specificity of anti-pSer18 in exclusively cross-reacting with 59 kD G6PD6 polypeptides phosphorylated at Ser18. The collective results confirm that in vivo G6PD6 phosphorylation at Ser18 occurs in *Arabidopsis* cell cultures and seedlings and is dependent on nutritional Pi status. It is also notable that Ser18 phosphorylation of G6PD6 in response to Pi resupply appears to be root- and cell-culture-specific as it was not detected on immunoblots of shoot extracts (Figure 6 and Appendix A). The anti-(pea G6PDc) immunoblots of extracts prepared from –Pi and/or Pi-resupplied cell culture and roots also detected an immunoreactive 70 kD polypeptide that did not cross-react with anti-pSer18 (Appendix A). This is likely due to a non-specific cross-reaction since the predicted size of *Arabidopsis* G6PD1-6 polypeptides ranges from 59 to 65 kD [4].

### 2.6. Pi-Resupply-Mediated Phosphorylation of Arabidopsis Root and Cell Culture G6PD6 at Ser18 Is Correlated with a Significant Increase in G6PD Activity and Biomass

G6PD activity was assayed in desalted protein extracts from –Pi and Pi-resupplied *Arabidopsis* seedlings (roots) and suspension cells cultivated in liquid nutrient media (Figure 7A,B). No contaminating ‘NADP^+^ reductase’ activity was detected in G6PD reaction mixtures lacking G6P. Furthermore, the addition of 2 mM dithiothreitol, which reduces disulfide bonds to dithiols [45], had no effect on G6PD activities of the root or cell culture extracts. This corroborates evidence that G6PD activity of heterotrophic plant tissues is largely cytosolic since G6PDp isozymes are reversibly inactivated following dithiothreitol-mediated disulfide-to-dithiol interconversion [1,2]. Pi resupply to –Pi seedlings correlated with a significant (54%) increase in the G6PD activity of root extracts (Figure 7B). Additionally, a significant 31% increase in root biomass was observed in the Pi-resupplied seedlings (Figure 7B). Likewise, Pi resupply induced a significant 27% and 39% increase in G6PD activity and biomass, respectively, of the *Arabidopsis* suspension cells (Figure 7B).

### 2.7. G6PD6 Phosphorylation at Ser18 Appears to Be Catalyzed by a CDPK

Semi-quantitative immunodot blot kinase assays using anti-pSer18 indicated that a CDPK present in clarified extracts of the Pi-resupplied *Arabidopsis* suspension cells catalyzes Ca^2+^-dependent phosphorylation of the synthetic G6PD6 deP-peptide shown in Figure 5A at Ser18 (Figure 8).

## 3. Discussion

G6PD is an integral enzyme of central C-metabolism that balances cellular redox status and maintains a critical control point for the diversion of G6P flux towards the OPPP for NADPH and pentose phosphate production (Figure 1). The present study presents evidence regarding the occurrence and possible function of multisite N-terminal phosphorylation of *Arabidopsis* G6PD6 in the context of Pi-nutrition. Sequence alignment and interrogation of phosphoproteomic datasets indicated that G6PD6’s N-terminal phosphosites are conserved in its paralogs and orthologs, and that G6PD6 and several of its orthologs from vascular plant and animal species are in vivo phosphorylated at similar sites in a range of tissues under various physiological conditions (Figure 2 and Table 1). It is notable that barley G6PDc (HvG6PD) and *Arabidopsis* G6PD6 were both phosphorylated at a comparable N-terminal serine residues following Pi resupply to –Pi plants (i.e., Ser14 and Ser13, respectively) (Figure 2 and Table 1), suggesting that this phosphorylation event may be a widespread G6PDc PTM of Pi-refed plants. By contrast, fungal and prokaryotic species have no conservation of *Arabidopsis* G6PD6’s N-terminal residues nor phosphosites (Figure 2). These results imply that N-terminal phosphorylation is a feature of most eukaryotic G6PDcs. Sequence analysis revealed that the flanking basophilic residues of G6PD6’s N-terminal phosphosites are reminiscent of motifs implicated in CDPK- and SnRK-mediated protein phosphorylation (Figure 3) [40,41,42]. Interestingly, peptide kinase assays using the synthetic G6PD6 deP-peptide (Figure 5A) as a substrate suggested that a CDPK phosphorylates G6PD6 at Ser18 in Pi-resupplied *Arabidopsis* (Figure 8). CDPKs catalyze site-specific phosphorylation of several regulatory enzymes of the central plant metabolism including nitrate reductase, sucrose synthase, sucrose phosphate synthase, and bacterial-type PEPC [41,46,47,48]. CDPKs also participate in nutrient signaling; for example, Liu et al. [49] found that *Arabidopsis* nitrate responses were regulated by CDPK-mediated phosphorylation, whereas Mehta et al. [18] reported that several CDPK isozymes and basophilic, CDPK-like motifs were enriched in the phosphoproteome of the Pi-resupplied *Arabidopsis* cell cultures.

Structural modeling and MobiDB interrogation indicated that the first 22 residues of G6PD6’s N-terminus correspond to an intrinsically disordered region. Annotation of the modeled G6PD6 structure (Figure 4) was based upon analysis of the human G6PD (HsG6PD) crystal structure, which revealed that the first 30 residues of its N-terminus are disordered [50]. Truncated HsG6PD mutants lacking these residues demonstrated normal catalytic activity, suggesting this region is neither essential for catalysis nor involved in tetramer/dimer formation [43,50]. Interestingly, Cys13 of HsG6PD forms a disulfide bridge with Cys446, which likely tethers down the mobile N-terminus to prevent interference with subunit association and catalysis. [50]. By contrast, *Arabidopsis* G6PD6 has no Cys residues within its N-terminal disordered domain (Figure 2) and would therefore lack the ability to form a stabilizing disulfide bridge. Disordered regions are frequently subject to site-specific phosphorylation and/or promote protein–protein interactions by providing a docking site for binding partners [51]. For example, castor plant bacterial-type PEPC contains an intrinsically disordered domain that (i) mediates its interaction with plant-type PEPC to form class-2 PEPC complexes that associate with the mitochondrial surface [52], and (ii) contains Ser425 and Ser451 phosphorylation sites that inhibit bacterial-type PEPC subunit activity within the class-2 PEPC complex [53,54]. Multisite N-terminal phosphorylation of G6PD6 following Pi resupply to –Pi *Arabidopsis* [18] could potentially stabilize its N-terminal disordered region and/or mediate the association with a binding partner, thus directly or indirectly influencing catalytic activity. However, the possibility that 14-3-3 proteins bind to and thereby regulate the activity of phospho-G6PD6 is unlikely since none of its N-terminal phosphosites are representative of a 14-3-3 binding motif (i.e., RSXpSXP or RXXpSXP where X is any amino acid) (Figure 2) [55]. Given the variability in the N-terminal sequences of orthologous plant G6PDc isozymes (Figure 2), perhaps the precise locations of G6PD6’s N-terminal phosphosites are not so important. Instead, the collective impact of multisite N-terminal phosphorylation would impart a pronounced, local negative charge that could potentially trigger downstream structural changes.

Phosphosite-specific antibodies provide a remarkably sensitive and specific immunochemical tool to visualize and quantify alterations in the phosphorylation status of a particular amino acid residue within a full-length protein sequence [56]. Immunoblotting using anti-pSer18 and anti-(pea G6PDc) antisera, respectively, revealed that Pi-resupply-mediated Ser18 phosphorylation of 59 kD G6PD6 polypeptides occurs in both *Arabidopsis* suspension cells and roots, with no obvious change in the amount of G6PD6 (Figure 6). These results corroborate those of Mehta et al. [18] who determined that although G6PD6 was in vivo phosphorylated at Ser18 (as well as Ser12 and Thr13) following Pi resupply to –Pi *Arabidopsis* suspension cells, its protein abundance was unaffected. The absence of G6PD6 Ser18 phosphorylation in the shoots of Pi-resupplied seedlings (Figure 6) implies that this PTM may play a greater role during Pi stress recovery in roots. It is important to note that the anti-pSer18 could potentially cross-react with pSer18 of G6PD5, as only three G6PD5 residues in positions 10–24 differ from the corresponding residues of G6PD6 (Figure 2). However, the occurrence of pSer18 in G6PD5 has rarely been documented (Table 1), and no change in G6PD5 phosphorylation nor protein abundance was detected following Pi resupply to the –Pi *Arabidopsis* cell cultures [18]. Furthermore, the cross-reaction of our anti-pSer18 antiserum with a Ser18 phosphorylated G6PD5 is unlikely since phosphosite-specific antibodies are capable of discriminating single-amino-acid substitutions occurring between the sequence of residues flanking phosphosites of paralogous or orthologous proteins [56].

Pi resupply significantly enhanced the G6PD activity of *Arabidopsis* roots and cell culture extracts (Figure 7B). This increase appears to occur post-translationally since G6PD6 protein abundance appeared to be unaffected between the two treatments (Figure 6) [18]. Similarly, Wakao et al. [12] concluded that the compensatory increase in G6PD6 activity of *atg6pd5* T-DNA mutants was likely due to a post-translational mechanism since *G6PD6* gene expression was unaffected in the *atg6pd5* mutants. The Pi-resupply-mediated increase in root and cell culture G6PD activity and biomass paralleled in vivo phosphorylation of 59 kD G6PD6 polypeptides at Ser18 (Figure 6 and Figure 7), suggesting this PTM helps to activate G6PD6. This novel mechanism of post-translational G6PD6 control is hypothesized to contribute to the activation of cytosolic OPPP flux, and thus the production of NADPH and C-skeletons (e.g., ribose-5-phosphate) needed to support the rapid resumption of cell growth that ensures Pi resupply to –Pi plants [14]. In this regard, the steady-state metabolic flux analysis carried out by Masakapalli et al. [17] was employed to define the metabolic phenotype of heterotrophic *Arabidopsis* suspension cells cultured in liquid nutrient media containing 0, 1.25, or 5 mM Pi. Fluxes through the central metabolic network were deduced from the redistribution of ^13^C into metabolic intermediates and end-products when the cells were labeled with ^13^C-glucose. Their results indicated that G6P flux through the cytosolic OPPP increased in parallel with the levels of Pi in the media. Moreover, Pi starvation triggered a complete cessation of the cytosolic OPPP, thus redirecting metabolic flux away from biomass accumulation towards glycolytic conversion of hexose to organic acid anions for Pi-conserving and Pi-scavenging pathways (e.g., PEPC, malic enzyme, etc.) [14,17].

It is interesting to note that the involvement of N-terminal phosphorylation in fine-tuning catalytic activity is a feature of several other regulatory branchpoint enzymes of the central plant C-metabolism. For example, peach leaf aldose-6-phosphate reductase, a dehydrogenase that converts G6P into glucitol (thus serving as a critical control point of sugar–alcohol metabolism), undergoes inhibitory N-terminal phosphorylation, hypothesized to coordinate sucrose and glucitol production during the diel cycle [57]. Pi deprivation of *Arabidopsis* cell cultures and seedlings triggered in vivo phosphorylation of the plant-type PEPC isozyme AtPPC1 at Ser11, which enhanced PEPC activity [15]. Similarly, sucrose synthase is phosphorylated at Ser11 during castor oil seed development, and sucrose phosphate synthase activity is modulated by N-terminal phosphorylation under cold and osmotic stress, as well as during the light–dark transition [58,59].

During salinity stress, *Arabidopsis* G6PD6 was reported to be phosphorylated by ASKα (a glycogen synthase kinase3 ortholog) at a conserved C-terminal threonine residue (Thr467) that is proximal to the structural NADP^+^-binding domain (Figure 4) [8]. This phosphorylation event was suggested to activate G6PD6 to maintain cellular redox balance and regulate stress tolerance; *askα* mutants were hypersensitive to salt stress as demonstrated by their increased ROS levels and impaired G6PD activity, whereas plants overexpressing *ASKα* exhibited the reverse phenotype [8]. This same phosphorylation event also appears to occur during *Arabidopsis* immune signaling: *g6pd6* mutants were more susceptible to *Pseudomonas syringae* infection than wild-type plants; this was reversed in *g6pd6* mutants complemented with T467D phosphomimetic, but not phosphoablative T467A G6PD6 mutants [60]. However, to the best of our knowledge, phosphorylation of *Arabidopsis* G6PD6 (or any of its orthologs) at Thr467 has not yet been corroborated in any plant phosphoproteomic study, including the *Arabidopsis* Pi resupply analyses of Mehta et al. [18]. Nevertheless, the possible involvement of Thr467 phosphorylation in post-translational G6PD6 control [8] in response to Pi nutrition should also be considered.

Unfortunately, our attempts to purify native G6PD6 from *Arabidopsis* cell cultures for detailed assessment of the impact of reversible phosphorylation on its kinetic properties have been hampered by the enzyme’s exceptional instability, resulting in unacceptably poor yields. To circumvent this issue, the kinetic and structural properties of recombinant wild-type G6PD6 [4] with corresponding site-directed phosphomimetic G6PD6 mutants (i.e., S3D, S12D, T13D and S18D) could be compared. This would not only ensure high-yield preparations, but also guarantee the purity of the enzyme for more definitive conclusions to be drawn. Furthermore, targeted LC-MS/MS analyses would provide a valuable indication of tissue- and site-specific changes in the phosphorylation status of G6PD6 and other proteins following Pi resupply to –Pi *Arabidopsis* seedlings. It will also be of considerable interest to identify the protein kinases and upstream signaling pathways responsible for in vivo multisite phosphorylation of G6PD6 in response to Pi nutrition. Mehta et al. [18] reported that (i) a total of 70 different protein kinases were differentially phosphorylated following Pi resupply to the –Pi *Arabidopsis* cell cultures (including CDPKs, mitogen-activated protein kinases, receptor-like cytoplasmic kinases and leucine-rich repeat receptor-like kinases), and (ii) CDPK-like motifs were enriched in the phosphoproteome of the Pi-resupplied cells. Our motif analysis of G6PD6 N-terminal phosphosites (Figure 3), and Ser18 peptide kinase assays (Figure 8) are consistent with the results of Mehta et al. [18] indicating the likely contribution of CDPKs in phosphoproteome remodeling in response to Pi nutrition.

## 4. Materials and Methods

### 4.1. Plant Material

Heterotrophic *Arabidopsis* (cv. Landsburg erecta) suspension cells were maintained at 21 °C in the dark as previously described [61]. Routine subculturing was performed by inoculating 40 mL of sterile Murashige and Skoog media [62] containing 1.25 mM Pi with 10 mL of a 7 d old culture. Cells used for time-course studies were obtained by subculturing 10 mL of a 5 d old culture into 40 mL of media containing 0 mM Pi (–Pi). Five days later, the flasks were supplemented with 2 mM Pi or maintained under –Pi; then, 48 h following Pi resupply, the cells were harvested via vacuum filtration through Whatman 541 filter paper, rinsed with ultrapure water, frozen in liquid N_2_ and stored at −80 °C. For liquid seedling cultures, 5 mg of seeds was surface sterilized via rinsing in 70% EtOH for 2 min, 50% bleach for 10 min; then, the seeds were thoroughly washed and suspended in sterile ultrapure water. Sterile seeds were stratified at 4 °C in the dark for 2 d, then placed in 250 mL Magenta boxes containing 50 mL of sterile 0.5x Murashige and Skoog medium (pH 5.7) with 1% (*w*/*v*) sucrose and 0.2 mM Pi. Seedling cultures were then placed on a New Brunswick Platform Shaker (model C10; Edison, CT, USA) at 80 rpm and 23 °C under continuous illumination (100 μmol m^−2^ s^−1^) (Figure 7A). After 7 d, the seedlings were transferred into 50 mL of fresh –Pi media and cultivated for an additional 5 d, at which point they were either supplemented with 2 mM Pi or maintained under –Pi; 48 h later, the roots and shoots were rapidly separated, weighed, frozen in liquid N_2_, and lyophilized.

### 4.2. Preparation of Clarified Protein Extracts

Tissues were ground to a powder under liquid N_2_ and then homogenized (1:2 (*w*/*v*) for quick-frozen suspension cells; 1:40 (*w*/*v*) for lyophilized shoots or roots) using a mortar and pestle with a small scoop of sand in 50 mM HEPES-KOH (pH 7.5), containing 10 mM MgCl_2_, 1 mM EDTA, 10% (*v*/*v*) glycerol, phosphatase inhibitor cocktail (20 mM NaF, 1 mM NaMoO_4_, 1 mM Na_3_VO_4_), 1 mM phenylmethylsulfonyl fluoride, 2% (*w*/*v*) poly(vinylpolypyrrolidone), and 0.02 mM NADP^+^. Homogenates were centrifuged at 14,000× *g* and 4 °C for 10 min. Clarified extracts (0.5 mL) were rapidly desalted at 400× *g* on 3 mL G-Biosciences (Saint Louis, MO, USA) spin columns containing 2.5 mL of Sephadex G-25 pre-equilibrated in extraction buffer lacking poly(vinylpolypyrrolidone) [63], then prepared for SDS/PAGE and immunoblotting, or protein and G6PD activity assays. 

### 4.3. G6PD Activity Assays and Protein Concentration Determination

G6PD activity was assayed at 23 °C by continuously monitoring the reduction of NADP^+^ to NADPH at 340 nm using a Spectramax Plus Microplate reader (Molecular Devices, San Jose, CA, USA). The optimized assay conditions were as follows: 50 mM Tris-HCl (pH 8.1) containing 20% (*v*/*v*) glycerol, 10 mM MgCl_2_, 0.2 mM NADP^+^ and either 0.5 mM 6PG plus 2.5 mM G6P, or 0.5 mM 6PG in a final reaction volume of 0.2 mL. To correct for contaminating 6PG-dehydrogenase activity, the activity measured in the presence of 6PG was subtracted from that measured with G6P and 6PG. The activity was proportional to the assay time and amount of enzyme added. One unit (U) of G6PD activity is the amount of enzyme resulting in the production of 1 μmol of NADPH per min.

Protein concentrations were determined using a bicinchoninic acid assay [64] with bovine γ-globulin as the protein standard.

### 4.4. Preparation of Anti-(phosphoSer18-Specific AtG6PD6) Antibody

Rabbit antiserum against pSer18 of AtG6PD6 was prepared using a synthetic phosphorylated peptide (LifeTein, Somerset, NJ, USA) corresponding to residues 10 through 24 of AtG6PD6’s N-terminus, with a phosphoryl group at the target Ser18 residue, and an additional N-terminal Cys residue to facilitate conjugation to keyhole limpet hemocyanin (KLH). KLH-conjugated P-peptide (1 mg) was reconstituted in Pi-buffered saline and emulsified 1:1 in Titermax Gold adjuvant (ThermoFisher Scientific, Mississauga, ON, Canada), then subcutaneously injected (500 µg) into a rabbit. Booster injections (250 µg each) were administered at 4 and 7 weeks; then, 7 d following the final injection, blood was collected into Vacutainers (Beckton Dickinson, Mississauga, ON, Canada) via cardiac puncture. Blood was centrifuged at 1000× *g* to remove coagulated blood cells, and the anti-pSer18 immune serum was frozen in liquid N_2_ and stored at −80 °C. The Animal Use Protocol for the production of rabbit antibodies was reviewed and approved by the Queen’s University Animal Care Committee, and the procedure was performed by Queen’s University Animal Care Services following the guidelines and policies of the Canadian Council on Animal Care.

### 4.5. Electrophoresis and Immunoblotting

SDS/PAGE was performed using a Mini-PROTEAN 3 gel electrophoresis mini-gel apparatus (Bio-Rad, Mississauga, ON, Canada). Stacking and resolving gels were made up of 4% and 10% (*w*/*v*) acrylamide, respectively, and were run at 200 V for 55 min. Following PAGE, the gels were electroblotted onto poly(vinylidene) difluoride membranes for immunoblotting. Immunoblots probed with anti-(pea G6PDc) were blocked in 3% (*w*/*v*) skim milk powder, whereas 3% (*w*/*v*) gelatin from cold water fish skin (Millipore-Sigma, Toronto, ON, Canada) was used to block anti-pSer18 immunoblots. Rabbit anti-(pea G6PDc) and anti-pSer18 immune sera were diluted 1:1000 and 1:500, respectively, in tris-buffered saline (20 mM Tris (pH 7.5), 150 mM NaCl) with 0.1% (*v*/*v*) Tween-20, containing bovine serum albumin or gelatin from cold water fish skin (1% *w*/*v* each) for anti-(pea G6PDc) and anti-pSer18, respectively, and 0.05% (*w*/*v*) NaN_3_. Antigenic polypeptides were visualized chromogenically using an alkaline-phosphatase linked secondary antibody as previously described [61]. All immunoblots were replicated a minimum of two times (with representative results shown), and the intensity of immunoreactive polypeptides was proportional to the amount of protein loaded.

### 4.6. G6PD Peptide Kinase Assays

Semiquantitative immunodot blot assays were performed using the synthetic G6PD6 deP-peptide (Figure 5A) as substrate to test for G6PD kinase activity in *Arabidopsis* suspension cell extracts. Clarified protein extracts prepared from liquid N_2_-frozen, 48 h Pi-resupplied cells were desalted as described above. The standard G6PD6 kinase assay mixture contained 50 mM HEPES-KOH (pH 7.3), 10 mM MgCl_2_, 10% glycerol, 1 mM dithiothreitol, and 0.5 mM MgATP. To assess Ca^2+^-dependent kinase activity, the samples were incubated in the standard G6PD kinase assay mixture with the addition of 0.2 mM CaCl_2_ or 5 mM EGTA. Clarified extract (4 µL, equivalent to 36 µg protein) was incubated for 0 or 15 min at 30 °C in the respective G6PD kinase assay mixture, containing 0.4 mg/mL deP-peptide in a final volume of 10 µL. Aliquots (1 µL) were pipetted onto a nitrocellulose membrane and probed (after blocking as described above) using anti-(pea G6PDc) or anti-pSer18 (+10 μg/mL deP-peptide). This experiment was replicated 3 times, with representative results shown in Figure 8.

### 4.7. Bioinformatic Analyses

G6PD amino acid sequences were retrieved from the National Center for Biotechnology Information (https://www.ncbi.nlm.nih.gov/protein/, accessed on 22 August 2023), and the protein BLAST tool was used to identify *Arabidopsis* G6PD6 orthologs. Multiple sequence alignments were executed using ClustalW version 2.1 (https://www.genome.jp/tools-bin/clustalw, accessed on 22 August 2023). The Eukaryotic Phosphorylation Site Database [65] (http://epsd.biocuckoo.cn/index.php, accessed on 22 August 2023) and Plant PTM Viewer version 2.0 [66] (https://bioinformatics.psb.ugent.be/webtools/ptm-viewer//, accessed on 22 August 2023) were used to identify experimental phosphorylation sites. The in silico predicted structure of AtG6PD6 was obtained from AlphaFold [67] (https://alphafold.ebi.ac.uk/, accessed on 15 August 2023) (model = AF-Q9FJI5-F1), and the residues were manually annotated in the SWISS-MODEL workspace [68] (https://swissmodel.expasy.org/, accessed on 15 August 2023) based on pairwise alignment with human G6PD using EMBOSS Matcher (https://www.ebi.ac.uk/Tools/psa/emboss_matcher/, accessed on 15 August 2023). The coenzyme and β+α domains (IPR036291 and IPR022675, respectively) were classified using InterPro [69] (https://www.ebi.ac.uk/interpro/, accessed on 15 August 2023). MobiDB (https://mobidb.bio.unipd.it/, accessed on 15 August 2023) was used to predict disordered regions [70].

### 4.8. Statistics

Unless stated otherwise, all statistical analyses were performed on a minimum of *n* = 3 biological replicates using Student’s *t*-test (two-tailed; unpaired) in GraphPad Prism (version 9.0). Data are reported as means ± SEM; statistical differences were deemed significant at *p* < 0.05.

## 5. Conclusions

Results of the current study indicate that *Arabidopsis* G6PD6 and its orthologs may be post-translationally regulated by multisite phosphorylation within their N-terminal disordered region. In particular, in vivo G6PD6 phosphorylation at Ser18 correlated with a significant increase in extractable G6PD activity and biomass accumulation in both Pi-resupplied *Arabidopsis* cell cultures and seedlings (i.e., roots). This mechanism of post-translational G6PDc control is hypothesized to help activate the enzyme to enhance cytosolic OPPP flux needed for anabolism and cell growth following Pi resupply. Understanding the mechanisms of post-translational G6PDc control, particularly those driven by abiotic stressors (or recovery from such), is important for broadening our understanding of the OPPP’s role in plant carbohydrate metabolism and redox homeostasis. Such research is foundational for the development of biotechnological tools and sustainable crop varieties, including those designed to optimize Pi utilization and acquisition, thus limiting our overuse of unsustainable and polluting Pi fertilizers in agricultural production. 

## Figures and Tables

**Figure 1 plants-13-00031-f001:**
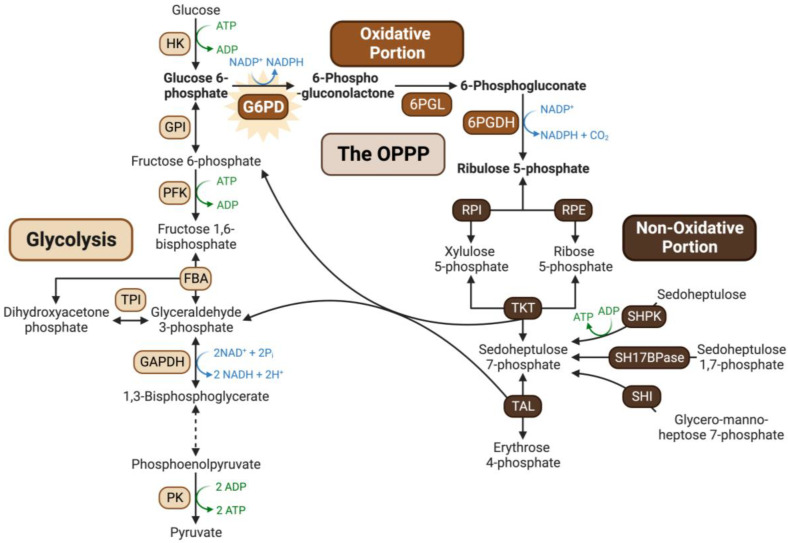
Overview of the oxidative pentose phosphate pathway. The OPPP integrates with the upper portion of the glycolytic pathway, providing a bypass for G6P oxidation to triose-phosphates while generating reducing power in the form of NADPH, and anabolic precursors such as ribose-5-phosphate and erythrose-4-phosphate required for nucleotide synthesis and the shikimate pathway, respectively. Abbreviations: 6PGL, 6-phosphogluconolactonase; 6PGDH, 6-phosphogluconate dehydrogenase; RPI, ribose 5-phosphate isomerase; RPE, ribulose 5-phosphate epimerase; TKT, transketolase; TAL, transaldolase; SHI, sedoheptulose 7-phosphate isomerase; SH17BPase, sedoheptulose 1,7-biphosphatase; SHPK, sedoheptulokinase; HK, hexokinase; GPI, glucose phosphate isomerase; PFK, phosphofructokinase; FBA, fructose biphosphate aldolase; TPI, triosephosphate isomerase; GAPDH, glyceraldehyde 3-phosphate dehydrogenase; PK, pyruvate kinase. Figure created with Biorender.com.

**Figure 2 plants-13-00031-f002:**
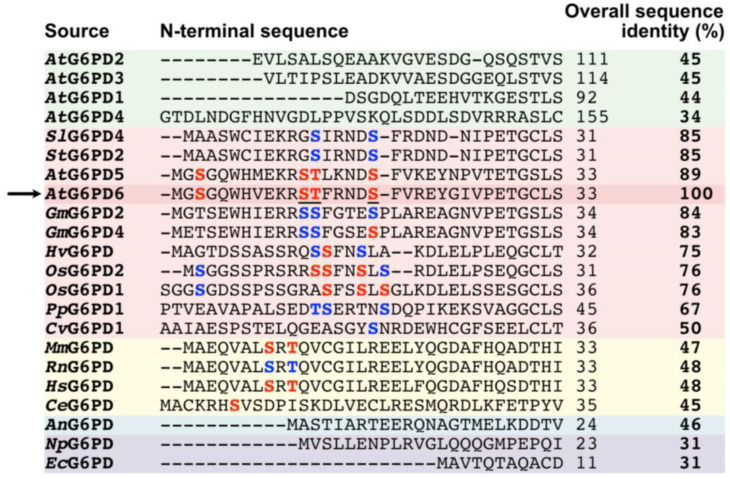
Multiple sequence alignment of *Arabidopsis* G6PD6’s N-terminus with N-termini of other G6PDs. G6PD6 phosphosites that were mapped via LC-MS/MS following 48 h of Pi resupply to –Pi *Arabidopsis* cell cultures [18] are colored in red and underlined, other experimentally determined phosphosites (Table 1) are colored in red, and corresponding conserved residues within other G6PDs are colored in blue. *Arabidopsis* G6PD6 is marked with an arrow. N-termini sequences are highlighted as follows: green = *Arabidopsis* G6PDp; pink = plant G6PDc; yellow = animal G6PD; blue = fungal G6PD; purple = bacterial G6PD. The abbreviations are as follows: An, *Aspergillus niger* (common mold); At, *Arabidopsis thaliana* (thale cress); Ce, *Caenorhabditis elegans* (nematode); Cv, *Chlorella vulgaris* (green alga); Ec, *Escherichia coli*; Gm, *Glycine max* (soybean); Hs, *Homo sapiens* (human); Hv, *Hordeum vulgare* (barley); Mm, *Mus musculus* (mouse); Np, *Nostoc punctiforme* (cyanobacterium); Os, *Oryza sativa* (rice); Pp, *Physcomitrium patens* (earthmoss); Rn, *Rattus norvegicus* (rat); Sl, *Solanum lycopersicum* (tomato); St, *Solanum tuberosum* (potato). Protein accession numbers are listed in Appendix A.

**Figure 3 plants-13-00031-f003:**
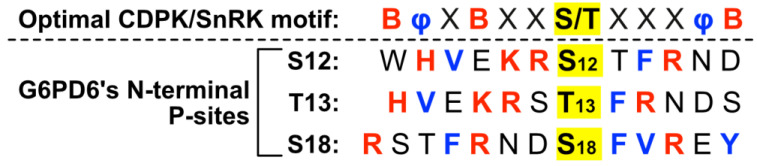
Motif analysis of phosphorylated *Arabidopsis* G6PD6 residues. Sequence alignment of amino acids flanking G6PD6 Ser12, Thr13, and Ser18 residues that were phosphorylated 48 h following Pi resupply to –Pi *Arabidopsis* cell cultures [18]; basic and hydrophobic residues are colored red and blue, respectively. The optimal CDPK/SnRK recognition motif is located above the alignment to highlight residues of interest, where φ represents a hydrophobic amino acid, B represents a basic amino acid, and X represents any amino acid.

**Figure 4 plants-13-00031-f004:**
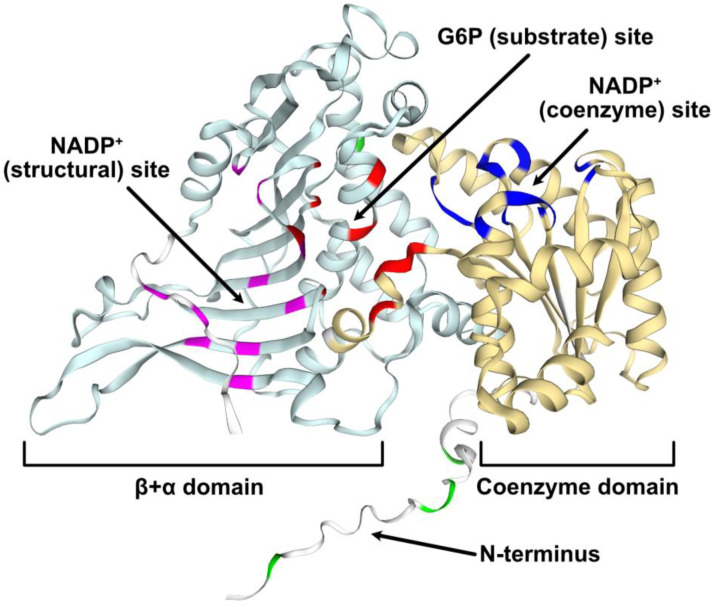
Structural model of *Arabidopsis* G6PD6. The predicted G6PD6 protein model (AF-Q9FJI5-F1) was obtained from AlphaFold and annotated in the SWISS-MODEL workspace. Based on pairwise sequence alignment with human G6PD, conserved key residues that interact with G6P or NADP^+^ (deduced from crystallized structures of human G6PD) are annotated as follows: red = G6P (substrate); dark blue = NADP^+^ (coenzyme); magenta = NADP^+^ (structural) [43]. Phosphorylated residues are annotated in green. Domains were retrieved from InterPro (coenzyme (yellow) = IPR036291, β+α (light blue) = IPR022675) and binding sites are annotated accordingly.

**Figure 5 plants-13-00031-f005:**
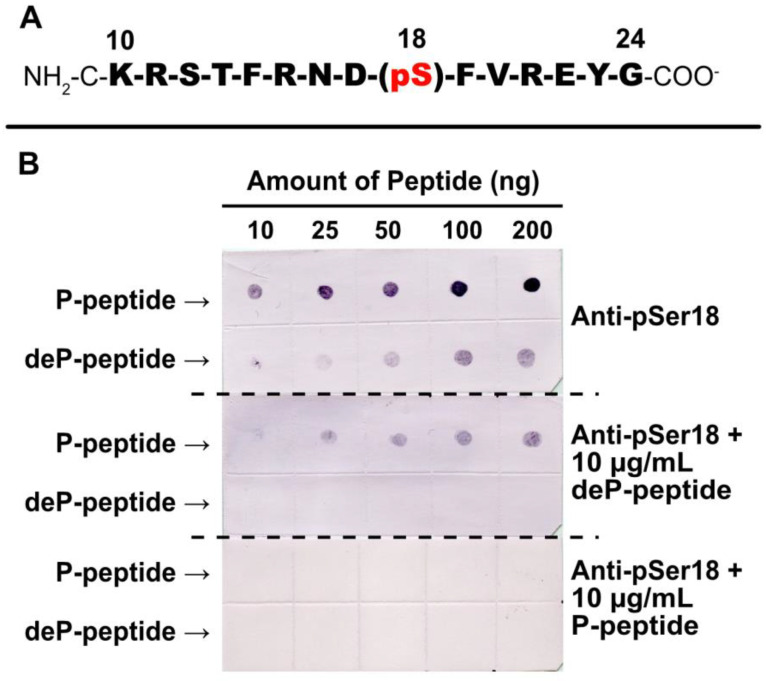
Specificity of phosphosite-specific antibody raised against pSer18 of *Arabidopsis* G6PD6. (**A**) Sequence of phosphorylated synthetic peptide that was covalently coupled to keyhole limpet hemocyanin (KLH) and used for rabbit immunization. The sequence numbering represents amino acid position relative to G6PD6’s N-terminus. The Ser18 phosphorylation site is indicated. The peptide was synthesized with an extra N-terminal Cys residue to facilitate its conjugation to KLH. (**B**) Anti-pSer18 immune serum (±10 μg/mL of phospho- (P-) or dephospho- (deP-) peptide) was used to probe immunodot blots of varying amounts of the pSer18 P-peptide and corresponding deP-peptide.

**Figure 6 plants-13-00031-f006:**
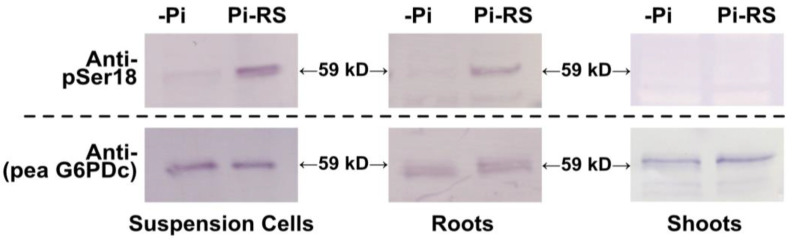
G6PD6 is in vivo phosphorylated at Ser18, 48 h following Pi resupply to Pi-starved *Arabidopsis* suspension cells and seedlings. Clarified extracts from –Pi and 48 h Pi-resupplied (Pi-RS) cells or seedlings (shoots and roots) were subjected to SDS/PAGE followed by immunoblotting with anti-pSer18 (+10 μg/mL dephospho- (deP)-peptide) or anti-(pea G6PDc). Approximately 8 and 1 µg of protein were loaded into each lane of the anti-pSer18 and anti-(pea G6PDc) immunoblots, respectively.

**Figure 7 plants-13-00031-f007:**
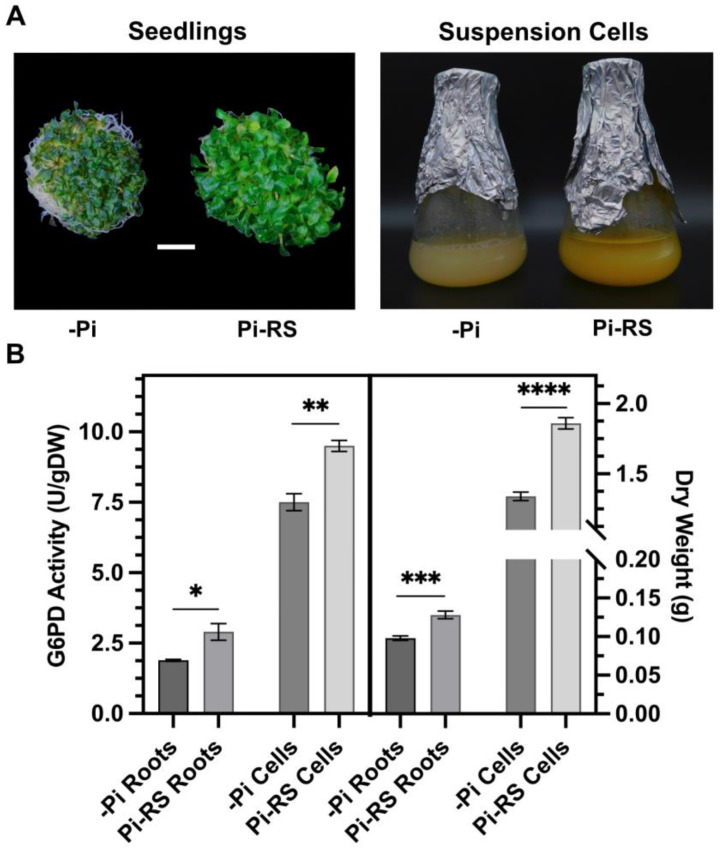
Impact of Pi deprivation (–Pi) and 48 h Pi resupply (Pi-RS) on appearance, G6PD activity, and biomass accumulation of *Arabidopsis* seedling roots and suspension cells. (**A**) Images are representative of at least five replicates (bar = 1 cm). (**B)** Dry weight values represent average gDW per flask of liquid cultured seedlings (roots) or suspension cells. All values represent means (±SE) of *n* ≥ 3 biological replicates; statistical significance was evaluated via a two-tailed, unpaired Student’s *t*-test (* *p* < 0.05, ** *p* < 0.01, *** *p* < 0.001, **** *p* < 0.0001).

**Figure 8 plants-13-00031-f008:**
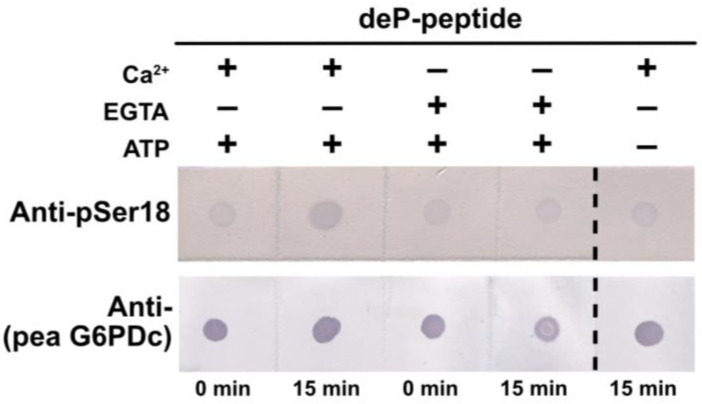
Evidence that a CDPK catalyzes G6PD6 phosphorylation at Ser18. Reactions were incubated at 30 °C for 15 min with (+) and without (−) 0.5 mM ATP, and either 0.2 mM Ca^2+^ (+) or 5 mM EGTA (−) in a final volume of 10 µL. The reactions containing 0.4 mg/mL of synthetic dephospho- (deP-) peptide (corresponding to residues 10–24 of G6PD6; Figure 5A) were incubated with 4 µL of desalted clarified extract (equivalent to 36 µg of protein) from Pi-resupplied *Arabidopsis* cell cultures, and then dot-blotted with anti-pSer18 (+10 μg/mL deP-peptide) or anti-(pea G6PDc) (400 ng peptide/dot).

**Table 1 plants-13-00031-t001:** Summary of vascular plant G6PDc phosphosites mapped during various phosphoproteomic studies.

Species	G6PDc Isozyme	Condition	Tissue	Phosphosites	Reference
** *Arabidopsis thaliana* **	G6PD6	Abscisic acid (ABA) application	Seedling	S3, T13, S18	[19,20]
		Development	Anther	T13, S18	[21]
		Auxin application	Root	S18	[22]
		Day-night transition	Rosette	S18	[23]
		Drought	Seedling	S3, T13, S18	[20,24]
		Ionizing radiation	Seedling	T13, S18	[25]
		Mannitol	Seedling	S18	[26]
		Pi resupply	Cell culture	S12, T13, S18	[18]
		Unstressed	Multiple	S3, S12, T13, S18	[27,28,29]
	G6PD5	Seed imbibition	Seed	S3	[30]
		Nitrogen re-supply	Seedling	S488	[31]
		Unstressed	Multiple	S3, S12, T13, S18	[27,32]
** *Glycine max* **	G6PD4	Aluminum stress	Root	S18	[33]
** *Hordeum vulgare* **	G6PD	Pi resupply	Root	S14, Y395	[34]
** *Oryza sativa* **	G6PD1	Brassinosteroid application	Seedling	S16, S19, S21	[35]
		Seed development	Seed	S21	[36]
			Multiple	S16, S21, Y501, T504, T513, S515	[37,38]
	G6PD2	ABA application	Seedling	S13	[39]
		Brassinosteroid application	Seedling	S13	[35]
		Seed development	Seed	S13	[36]
		Unstressed	Multiple	S12, S13, S16	[37,38]

## Data Availability

Publicly available datasets were analyzed in this study. All data generated and analyzed during this study are included in this article.

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
