# Peer review of "In Vivo Phosphorylation of the Cytosolic Glucose-6-Phosphate Dehydrogenase Isozyme G6PD6 in Phosphate-Resupplied Arabidopsis thaliana Suspension Cells and Seedlings"

_plants, 2023, doi:10.3390/plants13010031_

Round 1

Reviewer 1 Report

Comments and Suggestions for Authors

This article is related to the investigation of the phosphorylation of the N-terminal side of the AtG6PD6. Since few phosphorylation sites are known, S18 was selected for this investigation due to its higher phosphorylation rate variability. The present investigation is based on results obtained by immunodetection of the 18S site.

Despite the protein sequence used in the multiple alignment provided in Figure 2 seems to be convincing and few phosphorylation sites including the 12/13/18S are clear, I am questioning the absence of the 3S site. The multiple sequence alignment does not highlight the clear sequence homology between plants orthologue for the phosphorylated residue at position 12, 13 and 18 (and 3…). This figure must be improved to highlight the conserved pattern associated to multiple phosphorylation sites at the N-terminus of the G6PD6 isozime subgroup.

Questioningly, if mammalian species (human, mousse…) share similar protein N-termini as mention (page 4, lines 149-153) with the AtG6PD6, they must be included in figure 2.

Paragraph 2.2 need to be modified for a better understanding and annotated consensus sequence pattern or table must be preferred for clarity.

Considering the specificity of the anti-pSer18, Figure 4 clearly highlights the weak specificity (P/deP) of the antibody against the phosphorylated 18S since the dephosphorylated peptide is easily detected by the antibody (only 2.5 fold)… Moreover, the specificity of the antibody must have been tested against the other phosphorylation sites 3/12/13.

BTW, could the authors comment the advantage of this immunological approach since targeted LC-MS approach (MRM per example) will provide phosphorylation rate variation for all N-termini sites. This is especially relevant when the specificity (P/deP and potentially 2/12/13S sites) of the used antibody is not good enough to provide undoubtful results.

Then, although the investigation of the authors is an interesting starting study, results must be confirmed by mass spectrometry and to investigate the phosphorylation yield of the secondary sites.

To conclude, quantitative phosphoproteomics is required to consider the publication of the results.

Reviewer 2 Report

Comments and Suggestions for Authors

Dear Editor and authors: This interesting paper discusses the important of N-terminal phosphorylation of G6PD in the modulation of its catalytic activity in response to different phosphate availability. The authors also made a deep review about multiple phosphorylated residues in this domain and the physiological context and tissues where they have been detected. Although, previous phosphoproteomic analysis have showed that phosphate starvation conditions induce a multiple phosphorylation in the N-terminal of the protein, the interest was focused in the residue Ser18. They show that the level of phosphorylation of Ser18 is directly correlated with the level of G6PD activity and that a calcium dependent kinase is the responsible of this phosphorylation. Although there are not too much experimental results, they are novelty and open new possibilities for future assays. The manuscript is clearly written and well-structured and should be published. I have some questions about the manuscript that I would like discuss.

Pag. 8 Line 244, G6PD activity data must be showed in a chart or table to make easier the analysis. Why G6PD activity was not assayed in shoot in presence or not of DTT to confirm that the plastidic isoform is majority in shoot and it is not affected for phosphate nutrition.

Why do you conclude that a CDPK is the responsible of Ser18 phosphorylation? CDPK are directly regulated by Ca but is possible that a SnRK3, which is indirectly regulated by Ca and response to Pi starvation, could be related in this regulatory mechanism. To discern the putative calcium dependent kinase responsible of G6Pd phosphorylation, there are not any kinase between more than 500 proteins induce by Pi resupplied that your previous proteomic analysis detected.

In order to identified putative kinases involved in N-terminal phosphorylation, the previous result suggest me that, the fact that different level of N-terminal phosphorylation was found in different physiological context, several kinases could be related in the phosphorylation of the multiple phospho-target presented in the N-terminal domain in response to several physiological conditions.  

Pag 12 Line 421, How the Arabidopsis seedling were germinated, growth and treated is not clearly described? 

Pag 12 Line 427 Why the fresh tissue (shoot and roots) were lyophilized for prepared crude extract for protein assays?  I usually conserve the fresh tissue at -80ºC after frozen and homogenized with liquid N2.

Reviewer 3 Report

Comments and Suggestions for Authors

The manuscript plants-2727721 investigated in vivo phosphorylation of the cytosolic glucose-6-phosphate dehydrogenase isozyme G6PD6 in phosphate-resupplied Arabidopsis thaliana suspension cells and seedlings.

The contribution is interesting. However, methods need to be described to a greater extent. 

A picture of Arabidopsis thaliana suspension cells and seedlings used would be attractive.

Line 418. How was the seeds' surface sterilized and stratified?

Line 427. In 'lyophilized shoots or roots)' the parenthesis is not needed.

Line 446. 'Protein concentrations were determined using a bicinchoninic acid assay with bovine γ-globulin as the protein standard.' A reference is needed for the bicinchoninic acid assay.

Line 458. 'Rabbit inoculation and care was performed by Queen's University Animal Care Services.' Please indicate if an Animal Ethical Committee approved the experiment in agreement with the committee's guidelines for the control and supervision of animal experiments.

Please include details for all laboratory equipment (model, company, city, country).

The conclusion section must be improved; it is too general. A synthesis of critical points could highlight the primary results. It is also essential to recommend new areas for future research.

Round 2

Reviewer 3 Report

Comments and Suggestions for Authors

The authors of manuscript ID plants-2727721 'In vivo phosphorylation of the cytosolic glucose-6-phosphate dehydrogenase isozyme G6PD6 in phosphate-resupplied Arabidopsis thaliana suspension cells and seedlings' have improved the document.

One last concern: while there is no hard rule on length, conclusions are typically one paragraph long. The findings should restate your main points and provide closure. In the actual form, conclusions include references and mention figures; please modify.

Author Response

Response to Reviewer 3 Comments

  1. Summary

Thank you very much for taking the time to review our manuscript. Please find the detailed responses below and the corresponding revisions/corrections highlighted/in track changes in the re-submitted files.

Point-by-point response to Reviewer 3 Comments and Suggestions for Authors

Comments 1:   The authors of manuscript ID plants-2727721 'In vivo phosphorylation of the cytosolic glucose-6-phosphate dehydrogenase isozyme G6PD6 in phosphate-resupplied Arabidopsis thaliana suspension cells and seedlings' have improved the document. One last concern: while there is no hard rule on length, conclusions are typically one paragraph long. The findings should restate your main points and provide closure. In the actual form, conclusions include references and mention figures; please modify.

Response 1:  The manuscript has been revised in accordance with reviewer 3's valid comment. The 'Conclusions' section has been modified into a single, brief paragraph that restates our main points & provides closure. Several parts of the previous 'Conclusions' have been integrated into relevant portions of the Discussion. As a result we believe our  manuscript reads much better, and we are therefore grateful to this reviewer for their helpful comment.